# Constructing Multiple High-Quality Deep Neural Networks: A TRUST-TECH-Based Approach

## Abstract

The success of deep neural networks relied heavily on efficient stochastic gradient descent-like training methods. However, these methods are sensitive to initialization and hyper-parameters. In this paper, a systematical method for finding multiple high-quality local optimal deep neural networks from a single training session, using the TRUST-TECH (TRansformation Under Stability-reTaining Equilibria Characterization) method, is introduced. To realize effective TRUST-TECH searches to train deep neural networks on large datasets, a dynamic search paths (DSP) method is proposed to provide an improved search guidance in TRUST-TECH method. The proposed DSP-TT method is implemented such that the computation graph remains constant during the search process, with only minor GPU memory overhead and requires just one training session to obtain multiple local optimal solutions (LOS). To take advantage of these LOSs, we also propose an improved ensemble method. Experiments on image classification datasets show that our method improves the testing performance by a substantial margin. Specifically, our fully-trained DSP-TT ResNet ensmeble improves the SGD baseline by 15% (CIFAR10) and 13%(CIFAR100). Furthermore, our method shows several advantages over other ensembling methods.

## 1 Introduction

Due to the high redundancy on parameters of deep neural networks (DNN), the number of local optima is huge and can grow exponentially with the dimensionality of the parameter space (Auer et al. (1996); Choromanska et al. (2015); Dauphin et al. (2014b)). It still remains a challenging task to locate high-quality optimal solutions in the parameter space, where the model performs satisfying on both training and testing data. A popular metric for the quality of a local solution is to measure its generalization capability, which is commonly defined as the gap between the training and testing performances (LeCun et al. (2015)). For deep neural networks with high expressivity, the training error is near zero, so that it suffices to use the test error to represent the generalization gap. Generally, local solvers do not have the global vision of the parameter space, so there is no guarantee that starting from a random initialization can locate a high-quality local optimal solution. On the other hand, one can apply a non-local solver in the parameter space to find multiple optimal solutions and select the high-quality ones. Furthermore, one can improve the DNN performance by ensembling these high-quality solutions with high diversity.

TRUST-TECH plays an important role in achieving the above goal. In general, it computes high-quality optimal solutions for general nonlinear optimization problems, and the theoretical foundations can be bound in (Chiang & Chu (1996); Lee & Chiang (2004)). It helps local solvers escape from one local optimal solution (LOS) and search for other LOSs. It has been successfully applied in guiding the Expectation Maximization method to achieve higher performance (Reddy et al. (2008)), training ANNs (Chiang & Reddy (2007); Wang & Chiang (2011)), estimating finite mixture models (Reddy et al. (2008)), and solving optimal power flow problems (Chiang et al. (2009); Zhang & Chiang (2020)). Additionally, it does not interfere with existing local or global solvers, but cooperates with them. TRUST-TECH efficiently searches the neighboring subspace of the promising candidates for new LOSs in a tier-by-tier manner. Eventually, a set of high-quality LOSs can be found. The idea of TRUST-TECH method is the following: for a given loss surface of an op-

timization problem, each LOS has its own stability region. If one start from one local optimum, and track the loss values along a given direction, we will find an exit point where loss start to decrease steadily, which means another stability region corresponding to a nearby LOS is found. By following a trajectory in the stability region, another LOS is computed.

We propose an optima exploring algorithm designed for DNNs that is able to find high-quality local optima in a systematic way, and thereby form optimal and robust ensembles. Normally for a deep neural network, exit points can hardly be found by original TRUST-TECH due to the huge dimensionality. So, in this work we introduce the Dynamic Searching Paths (DSP) method instead of fixed directions. We set the search directions to be trainable parameters. After an exploration step forward along the current direction, we calibrate the direction using the current gradient. By doing so, the method can benefit from not only the mature Stochastic Gradient Descent (SGD) training paradigm with powerful GPU acceleration capability, but also exit points can be easily found.

The overall DSP-TT method consists of four stages. First, we train the network using local solvers to get a tier-0 local optimal solution. Second, our proposed Dynamic Search Path TRUST-TECH (DSP-TT) method is called to find nearby solutions in a tier-by-tier manner. Third, a selection process is performed so that candidates with high quality are chosen. Finally, ensembles are built with necessary fine-tunings on selected member networks. To the best of our knowledge, this paper is the first one to search for multiple solutions on deep neural networks in a systematical way.

Our major contributions and highlights are summarized as follows:

- We propose the Dynamic Searching Path (DSP) method that enables exploration on high-dimensional parameter space efficiently.

- We show that combining TRUST-TECH method with DSP (DSP-TT) is effective in finding multiple optimal solutions on deep neural networks systematically.

- We design and implement the algorithm efficiently that it obtains multiple local solutions within one training session with minor GPU memory overhead.

- We develop the DSP-TT Ensembles of solutions found by DSP-TT with high quality and diversity for further improving the DNN performance.

## 2 RELATED WORK

The synergy between massive numbers of parameters and nonlinear activations in deep neural networks leads to the existence of multiple LOSs trained on a specific dataset. Experiments show that different initializations lead to different solutions with various qualities (Dauphin et al. (2014a)). Even with the same initialization, the network can converge to different solutions depending on the loss function and the solver (Im et al. (2016)). Many regularization techniques are therefore proposed to force the network to converge to a better solution, some of which are proven to be useful and popular (Kingma & Ba (2015); Srivastava et al. (2014); Ioffe & Szegedy (2015)). However, it is still mysterious how these regularized solutions are compared to the global optimum.

There are researchers that focus on characterizing different local optima and investigating the internal relations among them. It is claimed in (Hochreiter & Schmidhuber (1997); Keskar et al. (2016)) that sharp minima prevent deep neural networks from generalizing well on the testing dataset. Later, Dinh et al. (2017) argued that the definition of flatness in (Keskar et al. (2016)) is problematic and came up with an example where solutions with different geometries can have similar test time performances. Li et al. (2018) designed a new visualization method that rebuilt the correspondence between the sharpness of the minimizer and the generalization capability. On the other hand, some researchers apply meta-heuristic algorithms to obtain a better local minimizer (Gudise & Venayagamoorthy (2003); Zhang et al. (2007); Juang (2004); Leung et al. (2003)). However, these methods were either designed for obsolete toy models or on explicit benchmark objective functions where there are analytical forms for global optimum, and therefore the effectiveness of these algorithms on deep architectures and large datasets seems unconvincing. Moreover, the advantage of the global searching ability seems to be crippled when it comes to deep neural networks, and the minimizers they found are still local. Recently, Garipov et al. (2018) reveal the relation among local optima by building pathways, called Mode Connectivities, as simple as polygonal chains or Bezier curves that

connect any two local optima. Draxler et al. (2018) also found similar results at the same time, although they used the Nudged Elastic Band (Jonsson et al. (1998)) method from quantum chemistry.

To address the issue of converging to suboptimal solutions, a great deal of research efforts were directed to ensembles. Xie et al. (2013) proposed horizontal and vertical ensembles that combine the output of networks at different training epochs. Laine & Aila (2016) used a group of models with different regularization and augmentation conditions to create variety. Moghimi et al. (2016) borrowed the concept of boosting to create a strong ensemble of CNNs. Izmailov et al. (2018) found that averaging weights from different iterations leads to flatter solutions than from SGD and helps in generalization. Huang et al. (2017a) proposed a method that obtains ensembles by collecting several local minima along a single training process using a cyclical learning rate schedule. Zhang et al. (2020) used similar approach, but with the sampling capability that fully exploits each mode. Garipov et al. (2018) developed the Fast Geometric Ensembling based on Mode Connectivity. Although the methods in these papers obtain multiple networks within one training session, these ensembles are still largely dependent on initialization. While these ensemble methods performs better than a single network, naive randomly initialized ensemble is still the best choice when training budget is unconstrained. Fort et al. (2020) explained this phenomenon as they explore different modes in function space compared to weight averaging. Shen et al. (2019) improved the ensemble inference efficiency via a teacher-student paradigm distilling the knowledge of an ensemble into one single network. Yang et al. (2020) built ensembles by randomly initialize on a subparameter space, aiming to alleviate the exponentially growing number of local minima on deep networks. Wang & Chiang (2011) used the TRUST-TECH (Chiang & Chu (1996); Lee & Chiang (2004); Chiang & Alberto (2015)) method to perform a systematic search for diversified minimizers to obtain their ensembles. They implemented TRUST-TECH for training and constructing high-quality ensembles of artificial neural networks and showed that their method consistently outperforms other training methods. We generalize this method and tailor it for deep architectures and to work efficiently with popular local solvers in deep learning.

## 3 TRUST-TECH METHOD FOR MULTIPLE OPTIMAL SOLUTIONS

### 3.1 TRUST-TECH METHODOLOGY

Another category of methods has been developed in recent years for systematically computing a set of local optimal solutions in a deterministic manner. This family of methods is termed TRUST-TECH methodology, standing for Transformation Under Stability-reTaining Equilibria Characterization. It is based on the following transformations:

(i) the transformation of a local optimal solution (LOS) of a nonlinear optimization problem into a stable equilibrium point (SEP, Chiang & Chu (1996)) of a continuous nonlinear dynamical system.

(ii) the transformation of the search space of nonlinear optimization problems into the union of the closure of stability regions of SEPs.

Hence, the optimization problem (i.e. the problem of finding LOSs) is transformed into the problem of finding SEPs, and therefore we use the terms LOS and SEP interchangeably in the following discussion. It will become clear that the stability regions of SEPs play an important role in finding these local optimal solutions. We note that, given a LOS, its corresponding first-tier LOSs are defined as those optimal solutions whose corresponding stability boundaries have a non-empty intersection with the stability boundary of the LOS (Chiang & Chu (1996); Lee & Chiang (2004)). The definition of the stability boundary and its characterization can be found in Chiang & Fekih-Ahmed (1996). Similarly, its second-tier LOSs are defined as those optimal solutions whose

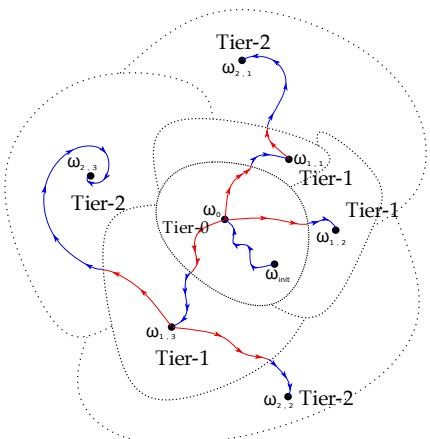

Figure 1: Given a LOS (i.e. $\omega_0$, a tier-zero LOS), the corresponding tier-1 LOSs are $\omega_{1,1}$, $\omega_{1,2}$, $\omega_{1,3}$. Similarly, its tier-2 LOSs are $\omega_{2,1}$, $\omega_{2,2}$, $\omega_{2,3}$. Note that the corresponding stability boundaries of tier-2 have a non-empty intersection with the stability boundaries of its tier-1 LOSs.

corresponding stability boundaries have a non-empty intersection with the stability boundary of first-tier LOSs (Chiang & Chu (1996); Lee & Chiang (2004)). See fig. 1 for an illustration.

We consider a general nonlinear unconstrained optimization problem defined as follows:

$$\min_x c(x) \tag{1}$$

where $c : D \subset \mathbb{R}^n \to \mathbb{R}$ is assumed to be continuously differentiable and $D$ the set of feasible points (or search space). A point $x^* \in D$ is called a local minimum if $c(x^*) \leq c(x)$ for all $x \in D$ with $\|x - x^*\| < \sigma$ for $\sigma > 0$.

To systematically search for multiple LOSs, a generalized negative gradient system based on the objective eq. (1) is constructed and is described by

$$\frac{dx}{dt} = -grad_R\, c(x) = -R(x)^{-1} \cdot \nabla c(x) = f(x(t)) \tag{2}$$

where the state vector $x(t)$ of this dynamic system belongs to the Euclidean space $\mathbb{R}^n$ and the function $f : \mathbb{R}^n \to \mathbb{R}^n$ satisfies the sufficient condition for the existence and the uniqueness of the solutions. $R(x)$ is a positive definite symmetric matrix (also known as the *Riemannian metric*) that generalizes various training algorithms. For example, if $R(x) = I$ (identity), it is a naive gradient descent algorithm. If $R(x) = \mathcal{J}(x)^\top \mathcal{J}(x)$ ($\mathcal{J}$ is the *Jacobian* matrix), then it is the Gauss-Newton method. If $R(x) = \mathcal{J}(x)^\top \mathcal{J}(x) + \mu I$, it becomes the *Levenberg-Marquardt* (LM) algorithm.

The *Theorem of the Equilibrium Points and Local Optima* (Lee & Chiang (2004)) shows one nice property of the gradient system (2), which is the critical point of the optimization problem (1) is a (asymptotically) SEP of the dynamic system (2). i.e. $\bar{x}$ is a SEP of (2) if and only if $\bar{x}$ is an isolated local minimum for (1). Hence, the task of finding the LOSs of (1) can be achieved by finding the corresponding SEPs of (2). In short, TRUST-TECH is a dynamical method designed to systematically compute multiple LOSs with the following features:

(i) it is a systematic and deterministic method to escape from a LOS towards another LOS,

(ii) it finds multiple LOSs in a tier-by-tier manner (see fig. 1), and

(iii) has a solid theoretical foundation (Chiang & Chu (1996); Lee & Chiang (2004); Chiang & Alberto (2015); Zhang & Chiang (2020)).

Another distinguishing feature of TRUST-TECH is its ability to guide a local method and/or a meta-heuristic method for effective computation of a set of LOSs or even the global optimal solution.

### 3.2 SYSTEMATIC SEARCH ON DEEP NEURAL NETS

Our method follows the paradigm of the TRUST-TECH. The center idea is to find multiple LOSs in a tier-by-tier manner. On small scale problems, applying fixed searching directions is proven to be effective in practice (Chiang et al. (2009); Reddy et al. (2008); Chiang & Reddy (2007); Wang & Chiang (2011)). In these applications, either random directions or eigen-vectors of the objective Hessian evaluated at each SEP were used. But in training deep neural networks, finding a proper direction is challenging. For a deep neural network, when searching along a random and fixed direction, the loss value will grow indefinitely. Another issue is that the computational cost of the original TRUST-TECH is high. Specifically, it assumes cheap evaluation of the objective function at each search step. However, in supervised learning of a large dataset, only the *empirical loss* is accessible instead of the ground-truth objective function. This means evaluating the loss function for the entire training set, which is almost impossible since it is limited by computational restrictions.

To tackle both challenges, we propose the Dynamic Search Path (DSP) method that enables exploration on deep neural networks' parameter space. Furthermore, we apply the DSP method to serve as the search paths for TRUST-TECH (DSP-TT). Details are discussed in section 3.2.1. An example of a one-tier DSP-TT method is shown in Algorithm 1.

### 3.2.1 OBTAINING DYNAMIC SEARCHING PATHS FOR TRUST-TECH

In this section, we go through the details on how to construct dynamic searching paths during TRUST-TECH computation and how it helps converge to nearby LOSs. Construction of searching path is inspired by the mode connectivity proposed in Garipov et al. (2018), in which the authors

---

**Algorithm 1** One-Tier DSP-TT Search

---
1: **procedure** T1SEARCH($model, dataset, maxiter, batchsize$)
2:     Initialize $paths, candidates = \{model.parameters\}$          ▷ intialize search paths and solution set.
3:     **for** $k \leftarrow 1$ to $maxiter$ **do**
4:         $batch \leftarrow$ getBatch($dataset, batchsize$)
5:         $\rho_1, \rho_2 \leftarrow$ Schedule($iter, maxiter, exit\_found$)          ▷ update learning rates
6:         $\Delta_k \leftarrow$ Select($paths$)          ▷ Randomly select one path
7:         Update($model, \Delta_k, \rho_1$)          ▷ forward search step
8:         $exit\_found \leftarrow$ CheckExit($model, \Delta_k, batch$)          ▷ check for exit on $k$th path
9:         **if** $exit\_found$ **then**
10:             $\omega_{s,k} \leftarrow$ LocalSolver($model, \Delta_k, dataset$)          ▷ converge to tier-1 optimum
11:             $candidates = candidates \bigcup \{\omega_{s,k}\}$          ▷ update solution set
12:         **else**
13:             Calibrate($model, \rho_2, \Delta_k, batch$)          ▷ calibration step
14:     **return** $candidates$

---

found there exists low-loss "tunnels" between different LOSs. But Mode Connectivity is used to find a high-accuracy pathway between two local solutions by optimizing the expectation over a uniform distribution on the path. Our focus is finding the proper searching directions towards nearby optima when starting from one LOS. They also claimed that a path $\phi_\theta$ cannot be explicitly learned when given one starting point. However, we find that such a construction is possible. Specifically, by redesigning the objective and combining the exploration capability of TRUST-TECH and the exploitation capability of SGD, another local optimum can be found starting from one LOS. More generally, by using such an optimization-based path-finding technique, one can find multiple tier-one SEPs (i.e. nearby LOSs) simultaneously.

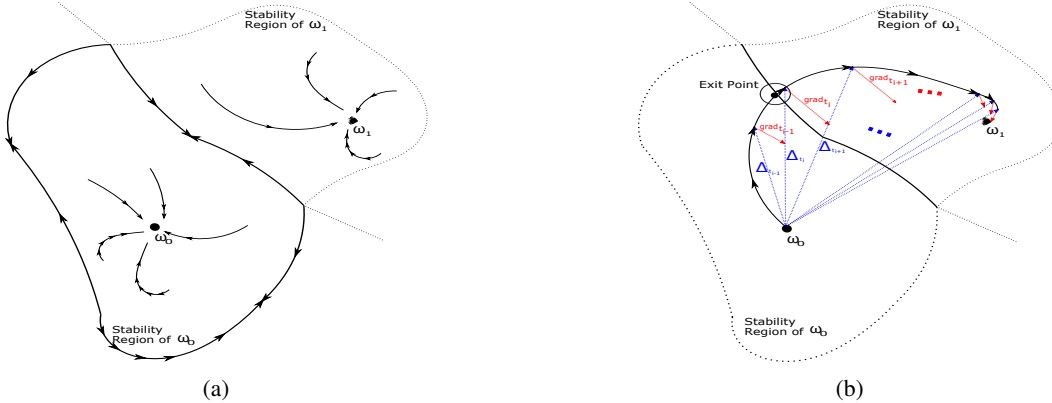

(a)          (b)

Figure 2: (a) Phase portrait of a two-dimensional objective surface, $\omega_0$ and $\omega_1$ are two SEPs; (b) A demonstration of the DSP method on the same objective. Black arrows represent the DSP path, blue vectors represent forward search steps, and red vectors represent calibration steps.

To do this, we first train the neural network to obtain a LOS $\omega_0 \in \mathbb{R}^{|net|}$. Then we define a trainable search direction vector $d_i \in \mathbb{R}^{|net|}$ (randomly initialized at $d_0$), so that during a TRUST-TECH search from $\omega_0$, instantaneous parameter vector at step $i$ is represented as $(\omega_0 + d_i)$.

At each step $i$, DSP updates the direction $d_i$ as:

$$d_i = \rho_1(i) \cdot d_{i-1} + \rho_2(i) \cdot f(\omega_0 + d_i) \tag{3}$$

The first term describes $\rho_1(i) \cdot d_{i-1}$ the original TRUST-TECH search with no direction calibration, where $\rho_1(i) \in (0, \rho_{max}]$ is the step size schedule for exploration phase whose value increases from 0 to $\rho_{max}$ w.r.t. step $i$. The second term is the DSP calibration term, where $\rho_2(t_i)$ is the step size schedule for the calibration, $descent_d$ represents a general local descent solvers, such as Gradient Descent, Newton's Method, etc. $f(\cdot)$ is the dynamics defined in Equation (2), where various local solvers can be applied here. The stopping criteria of $\rho_1(t_i)$ is determined dynamically by either an exit point is found or $\rho_{max}$ is reaches.

The above steps repeats until $(\omega_0 + d_i)$ converges to another LOS, which we call it a *tier-1 solution* associated with $\omega_0$. An intuitive demonstration of this process is shown in Figure 2b.

Our proposed scheme is scalable to performing multiple search directions starting from one LOS. To do this, we initialize multiple directions, and at each step, each search direction is updated via eq. (3). It is also worth noting that during training, the computation graph size is the same as the original network, since the algorithm only picks one direction to be included in the computation graph. Thus, minor memory overhead is introduced in practice. As for the computational efficiency, our proposed method evaluates objectives on mini-batches instead of the entire dataset, and determines the stopping criteria by an exponential moving average of past batch evaluations. To further stabilize the stochastic behavior caused by mini-batch evaluations, buffer variables are used to determine the state transition between up (loss values are climbing in current stability region) and down (reaches a nearby stability region and the loss decreases steadily). These resolve the efficiency issue of the original TRUST-TECH on large scale supervised learning problems.

## 4 DSP-TT ENSEMBLES OF DEEP NEURAL NETWORKS

When training budget is less constrained, *high-quality* of each tier-1 solution is emphasized as having better test accuracy than the tier-0 network. On the other hand, for building ensembles with a limited budget, *high-quality* is emphasized more on the diversity among the collection of local optimal neural networks found to better serve the ensemble, in stead of on a single network.

With the proposed DSP-TT, a set of optimal network parameters with high accuracy can be found systematically given enough training budget, and with limited budget, the high diversity among tier-0 and tier-1 solutions still remedies the weaker performance on tier-1 networks when serving the ensemble. Individual qualities are guaranteed because the starting point of any search is already a LOS from mature SGD-based solvers with high quality, which is also shown from the experiments, especially in Table 4. As for diversity, SEPs (i.e. optimal parameter values, or LOS) are separated by at least two stability regions because each SEP has its own stability region. It is necessary to initialize parameters in different stability regions in order to find multiple optimal solutions. The proposed TRUST-TECH based method is systematic in characterizing stability regions while other heuristic-based algorithms are not. And therefore, the diversity among SEPs found by our method is also high due to the mutual exclusiveness of stability regions.

The high-quality LOSs with high diversity further motivate us to build ensembles to make a more robust and accurate model than each single member. First, a list of candidates with high quality and diversity are selected. After that, a fine-tuning process is executed if necessary to help any under-fitted candidates toward greater convergence. Since the searching process already integrates the gradient information, the fine-tuning in our algorithm requires little effort. In fact, as shown in the experiments, fine-tuning does not show a benefit for the ensembling performance, so this procedure is ignored by default. Finally, we build the final ensembles by either averaging (regression) or voting for (classification) the outputs. Sophisticated ensembling methods can be applied here, however it is out of the scope of this paper.

## 5 EXPERIMENTS

Exit point verification is run using MLPs on UCI-wine and MNIST datasets. Further experiments are run using VGG-16 (Simonyan & Zisserman (2014)), DenseNet-100-BC (Huang et al. (2017b)) and ResNet-164 (He et al. (2016)) on CIFAR datasets. The program is developed on PyTorch framework. Each configuration is run multiple times and the average performance are shown.

**Hyperparameters** *Training budget:* DenseNet has 300 epochs of training budget, and ResNet/VGG has 200 epochs. *Batch size:* 128 for VGG and ResNet, and 64 for DenseNet. *DSP-TT parameters:* $\rho_1$ increases 0.001 per iteration, $\rho_2$ is $0.1\times$ of the initial tier-0 learning rate. Fine-tuning phase requires 10 epochs per solution. *All others:* DenseNet follows Huang et al. (2017b), VGG and ResNet follows Garipov et al. (2018).

For DSP-TT ensembles, exit points are usually found in negligible time (e.g. around 1min on CIFAR compared to a full training which takes hours). So 50 epochs are given to one tier of DSP-TT search with all exit points, while the rest of the budget are given to tier-0 training.

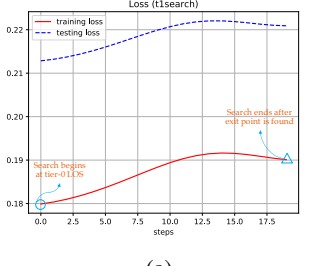 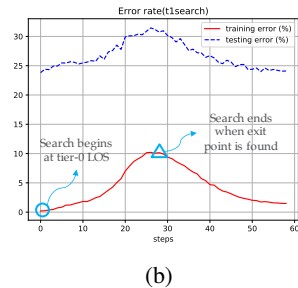 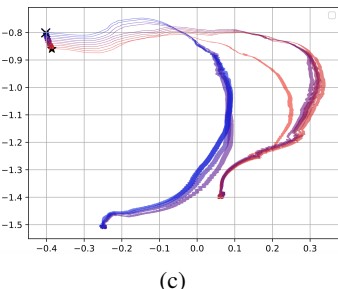

(a)          (b)          (c)

Figure 3: (a) Loss progress on the training and testing set of MNIST during a DSP-TT(full gradient) search. (b) Error rate progress on the training and testing set of CIFAR100 during a DSP-TT(batched) search. (c) Exit Point Verification: Points along the search path near the exit point (top left with "X" marker) are sampled and then integrated until convergence. The points before (red) and after (blue) exit converge to different LOSs.

## 5.1 EXIT POINT VERIFICATION

Exit points play an important role in TRUST-TECH method in finding multiple local optimal solutions. Figures 3a and 3b shows full gradient and batch version of a loss change with respect to the DSP-TT search iterations along one search path. The loss value first goes up, escaping from the tier-0 solution. At a certain point, the loss reaches a local maximum and then goes down, suggesting that the search path hits the stability boundary and enters a nearby stability region.

To further verify that an exit point lies on the stability boundary, we do the following visualization: Several points along the search path near the exit point are sampled. Then a forward integration (gradient descent with small step size) is executed starting from each sample. Trajectories are plotted by projecting the parameter space onto two random orthogonal directions. Due to high computation cost, this process is only simulated using a 1-layer MLP with 5 neurons (61 parameters) trained on UCI-wine dataset. Each integration process is executed for 50,000 steps with step size of 0.01. As shown in fig. 3c, The points before (red) and after (blue) exit converge to two different points on the 2D projection space. We also observe the cosine between the initial and updated search directions remains close to 1.0 throughout the search process, suggesting that gradients only calibrate extreme dimensions of the initial direction, but does not interfere with the remaining majority of dimensions.

## 5.2 TIER-BY-TIER SEARCH

The proposed DSP-TT computes: 5 tier-one (from tier-zero LOS) and 5 tier-two (from the best tier-one LOS) LOSs. Among these, we perform the following ensembles: *Tier-1* (5 tier-one LOSs); *Tier-1-tune* (5 tier-one LOSs, each with a fine-tuning); *Tier-0-1* (1 tier-zero and 5 tier-one LOSs); *Tier-0-1-2* (1 tier-zero, 5 tier-one and 5 tier-two LOSs). We use SGD as the local solver and DenseNet as the architecture. As shown in table 1, all DSP-TT-enhanced ensembles outperform the baseline model. Although *Tier-0-1-2* performs mostly best among all, it is sufficient to use *Tier-0-1* in practice for efficiency, and

|  | CIFAR10 | | CIFAR100 | |
|---|---|---|---|---|
|  | err (%) | loss | err (%) | loss |
| Baseline | 5.92* | / | 24.15* | / |
| Tier-1 | 4.65 | 0.1622 | 22.29 | 0.9014 |
| Tier-1-tune | 4.64 | 0.1670 | 22.29 | 0.9206 |
| Tier-0-1 | **4.34** | 0.1433 | 20.85 | 0.8323 |
| Tier-0-1-2 | 4.36 | **0.1396** | **20.56** | **0.8090** |

Table 1: Error rate and cross-entropy loss on the test set of CIFAR10 and CIFAR100 datasets with different tier settings. (Baseline: the DenseNet-100-BC trained by SGD. *: numbers from Huang et al. (2017b))

therefore we use *Tier-0-1* in all the following experiments. From table 1, we also find that although fine-tuning individuals can improve its own performance, it does not help much on the ensembles performance. This shows that the diversity introduced by our algorithm dominates the fine-tuning improvements by individuals. So in later experiments, all fine-tunings are neglected.

| Architecture(Budget) | Method(#models) | Test Error (%) | | Output Correlation | |
|---|---|---|---|---|---|
| | | C10 | C100 | C10 | C100 |
| VGG16(200) | Individual(1) | 6.75* | 27.4* | / | / |
| | SSE(5) (Huang et al. (2017a)) | 6.57* | 26.4* | / | / |
| | FGE(22) (Garipov et al. (2018)) | **6.48*** | 25.70* | / | / |
| | DSP-TT(6) | 6.56 | **25.64** | 0.976 | 0.957 |
| ResNet-164(200) | Individual(1) | 4.72 | 21.50 | / | / |
| | SSE(5) | 4.67 | 20.68 | 0.917 | 0.815 |
| | FGE(6) | 4.67 | 20.75 | 0.944 | 0.902 |
| | DSP-TT(6) | **4.53** | **20.20** | **0.898** | **0.773** |
| DenseNet-100(300) | Individual(1) | 5.92** | 24.15** | / | / |
| | SSE(6) | 4.48 | 21.08 | 0.946 | 0.901 |
| | FGE(6) | 4.82 | 22.82 | 0.992 | 0.991 |
| | DSP-TT(6) | **4.34** | **20.85** | **0.903** | **0.832** |

Table 2: Performance comparison among ensemble methods on various architectures on CIFAR datasets. (Output Correlation: mean Pearson Correlation Coefficients among all members. *: numbers from Garipov et al. (2018), **: numbers from Huang et al. (2017b). Source codes for FGE (https://github.com/timgaripov/dnn-mode-connectivity) and SSE (https://github.com/gaohuang/SnapshotEnsemble) are from the authors, respectively, and we present the best results we could achieve.)

| Method(#models; Budget) | Test Error (%) | | Parameter Distance | | Output Correlation | |
|---|---|---|---|---|---|---|
| | C10 | C100 | C10 | C100 | C10 | C100 |
| Individual(1;200) | 4.72 | 21.50 | / | / | / | / |
| Individual-Ens(6;1200) | 4.35 | 19.79 | 42.19 | 71.64 | 0.912 | 0.756 |
| SSE(6;200) (Huang et al. (2017a)) | 4.67 | 20.68 | 29.70 | 44.18 | 0.917 | 0.815 |
| FGE(6;200) (Garipov et al. (2018)) | 4.67 | 20.75 | 12.70 | 19.06 | 0.944 | 0.902 |
| DSP-TT-0-1(6;200) | **4.53** | **20.20** | 39.41 | 69.09 | 0.898 | 0.773 |
| DSP-TT-0-1-full-train(6;1200) | **3.99** | **18.67** | 35.82 | 82.11 | 0.929 | 0.810 |

Table 3: Detailed comparison with ResNet-164 trained on the CIFAR datasets. (Parameter Distance: Euclidean distance of parameters.)

## 5.3 COMPARISON WITH OTHER ENSEMBLE ALGORITHMS

In this section, we compare our method with other popular ensemble methods (Huang et al. (2017a); Garipov et al. (2018)) in deep learning. Results are shown in tables 2 and 3.

Besides accuracy, member diversity is another major quality for ensembles. Ideally, we want all members perform relatively well, while each member learns some knowledge that differs from that of others. We measure the output correlation (Huang et al. (2017a)) and the parameter distance (Garipov et al. (2018)). In table 2, the correlation by DSP-TT outperforms other ensemble methods. And a more detailed analysis in table 3 shows that both parameter distance and output correlation by DSP-TT Ensembles are better than SSE and FGE, and are at a similar level to those of Individual Ensembles (multiple networks trained from scratch). Moreover, our fully trained DSP-TT Ensembles outperforms Individual Ensembles, and improves the individual baseline by 15% (CIFAR10) and 13% (CIFAR100). table 4 shows that fully trained tier-1 networks performs at least as good as the tier-0 network. This suggests that training from an exit point found by DSP-TT method is better than from a random initialization. It is notable that in multiple cases, FGE members are more correlated, indicating that these members are not multiple LOSs, but perturbations near one LOS. From this perspective, FGE can be regarded as a fine-tuning around one local optimal point.

| | Test Error (%) | |
|---|---|---|
| | CIFAR-10 | CIFAR-100 |
| Tier-0 | 5.00 | 21.98 |
| Tier-1(#1) | 4.86 | 21.58 |
| Tier-1(#2) | 4.68 | 21.30 |
| Tier-1(#3) | 4.78 | 21.72 |
| Tier-1(#4) | 4.89 | 21.98 |
| Tier-1(#5) | 4.73 | 21.72 |

Table 4: Test error rate of individual networks of the DSP-TT-0-1-full-train Ensemble from table 3. All tier-1 networks performs at least as good as the tier-0 network.

From the hardware side, DSP-TT search process introduces minor overhead to the GPU memory usage. Specifically, baseline training of ResNet-164 takes 3819Mb GPU memory, which increases to 3921Mb during DSP-TT search. This justifies our previous claim that TRUST-TECH does not increase the size of the computation graph with only a little additional overhead.

## 5.4 ABLATION TEST ON DSP-TT HYPERPARAMETERS

The key hyperparameters for DSP-TT are $\rho_1$ (pace of search step) and $\rho_2$ (step size of calibration step) defined in Section 3.2.1. In this part we test the sensitivity of the two. We perform tests on a grid of $(\frac{d\rho_1}{dt}, \rho_2)$ pairs, and record (1) the number of iterations to finish a DSP-TT search for exit points, (2) average $\rho_1$ of each search path when an exit point is reached, and (3) average distance between the search origin (tier-0 solution) and each exit point. As shown in Figure 4, DSP-TT is insensitive to $\rho_2$. And figs. 4b and 4c show (1) $\rho_1$ and the distance between tier-0 and exit points are highly correlated, and (2) The surface becomes flat after the increment speed of $\rho_1$ passes $5e - 4$, suggesting that other stability regions are reached.

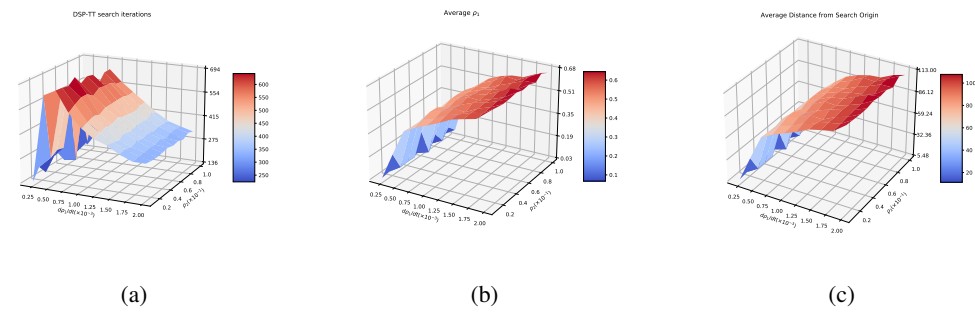

(a)   (b)   (c)

Figure 4: Sensitivity test on $\rho_1$ an $\rho_2$. X-axis: increment rate of $\rho_1$; Y-axis: $\rho_2$; Z-axis: (a) Running iterations when all exit points are reached. (b) Average $\rho_1$ when each exit point is reached. (c) Average distance from the search origin when each exit point is reached.

## 6 CONCLUSION AND FUTURE WORK

In this paper, we propose a novel Dynamic Search Path TRUST-TECH training method for deep neural nets. Unlike other global solvers, our proposed method efficiently explores the parameter space in a systematic way. To make the original TRUST-TECH applicable to deep neural networks, we first develop the Dynamic Searching Path (DSP) method. Second, we adopt the batch evaluation formula to increase the algorithm efficiency. Additionally, to further improve the model performance, we build the DSP-TT Ensembles. Test cases show that our proposed training method helps individual models obtain a better performance, even when tier-1 search is applied. Our method is general purposed, so that it can be applied to various architecture with various local solver.

Moreover, it is observed from Table 1 that percentage improvements in error rate is not as significant as that in loss. This suggests that the cross-entropy loss may be the bottleneck for further improvements in performance for classification tasks. Thus, designing a proper loss function that can be more sensitive to classification accuracy would be a valuable topic in the future.

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
