# OpenReview forum: "Constructing Multiple High-Quality Deep Neural Networks: A TRUST-TECH Based Approach"
_ICLR.cc/2021/Conference — Reject_

### Official Review · AnonReviewer4 · 2020-10-28
**Conceptually simple, but seemingly effective way of finding diverse local optima and building model ensembles**

**Rating:** 6
**Confidence:** 3

**Review:**

##########################################################################

Summary:

The paper describes a technique based on the modified generalized gradient descent for finding multiple high-quality local optima of deep neural networks. The search method does not require re-initialization of the model parameters and can be carried out in a single training session. Identified local optima are then used to build model ensembles which appear to outperform several other ensembling approaches.


##########################################################################

Reasons for score:


Overall, I vote for accepting. While I was not entirely satisfied with how the paper is written and how the approach is introduced and explained, I find the results to be quite interesting. While I do not know the literature sufficiently well, I think this method is original and well-founded.


##########################################################################

Pros:

1. While being simple and intuitive, the proposed method appears to succesfully and efficiently identify multiple high-quality local optima of a model.

2. Possibly even more interestingly, ensembles containing corresponding models appear to outperform other alternative approaches. While inference with ensembles of models can be quite costly (growing with the number of models in the ensemble) and a similar or better accuracy could potentially be achieved with larger simple models sharing the same computational cost, this result is nevertheless very promising.

3. The experimental methodology appears to be sound and some illustrative examples (like those shown in Figure 3) are interesting and insightful.


##########################################################################

Cons:

1. Certain parts of the publication are not entirely well written and some sentences are a bit confusing. Also, the text contains quite a few misprints. Some more serious mistakes can be found, for example, in Table 2 (DenseNet results) and central equations (5) and (6), which seem to use a wrong sign for the gradient (current sign seems to correspond to gradient ascent and not descent thus maximizing the loss and not minimizing it). Also, as a very minor comment, I believe that, strictly speaking, the gradient (with respect to $\Delta$) of the loss in these equations should be computed at $\phi_{\omega_0}(t_{i-1})$ because otherwise the right-hand side of these equations would be dependent on $\Delta(t_{i})$.

2. In my opinion, the discussion in Section 3 could be clarified and simplified. Furthermore, I believe that the method could be explained and analyzed a bit better. For example, it would appear that the proposed difference equation (6) can be written as a gradient descent on the modified loss function with the added quadratic term $\sim (\rho_1/\rho_2) \Delta^2$. If correct, I find this simple perspective much more natural and insightful. This quadratic component can essentially flip the Hessian in the vicinity of the starting local minimum thus causing the trajectory to be repelled from it. This simple view also appears to have implications for what kind of local minima of the original loss (their Hessians) could finally attract such training trajectories, potential shifts due to finite $\rho_1$, and the role that the decay of $\rho_1/\rho_2$ could play in the process of convergence.

3. The related work section contains just a few ensemble papers and none after 2018. It would appear that this section could be expanded and include some more recent papers at least for reference.

##########################################################################

Questions during rebuttal period:

Please address and clarify the cons above. (I will update my score based on the authors reply.)

Unfortunately, I am not an expert in this field, but two papers I came across doing a very quick search appear to be somewhat relevant: "Local minima found in the subparameter space can be effective for ensembles of deep convolutional neural networks" and "MEAL: Multi-Model Ensemble via Adversarial Learning". I am not sure these particular papers need to be included in the prior work section, but I do think that this publication would benefit from a more in-depth literature overview.

##########################################################################

Post-rebuttal.

Thanks for a detailed response that clarified some of my questions. I think the overall quality of the paper increased and I am happy to see additional information (like additional literature and an ablation study in Section 5.4) and a somewhat improved explanation of the core idea. However, I am still hesitant to give this paper a higher rating, in part because I find Section 5.4 to be poorly written and somewhat confusing (it lacks any sort of conclusion or insight and I cannot read Figure 4 at all, the text is too small) and partly because the paper does not clearly put its results in the context of the recent model ensemble progress (and thus makes me doubt the impact of this result on the field; although it does seem to be promising compared to the mentioned baselines).

---

> ### Author Response · Authors · 2020-11-18
> **Response to Reviewer 4**
>
> We appreciate your insightful comments and feedback. We have made changes in the updated manuscript. And the followings are specifically addressing your concerns.
> - We realized the type in Table 2 right after we submitted the manuscript, and it should be fixed in the new version. (Specifically, the CIFAR100 test error for FGE on DenseNet100 should be 22.82 instead of 0.992.)
> - For equations (5) and (6) (Equation 3 in updated version), $grad_{\Delta}$ should be defined as negative gradients for minimization problems and positive gradients for maximization problems. In the updated manuscript, we modified Section 3.2.1 trying to make the derivation more concise.
> - We appreciate your insightful analysis on the DSP-TT algorithm. Following your suggestion, we modified Section 3.2.1 with more subtle notations. We agree that the original Equation 6 (new equation 3) can be understood as adding (for minimization, should be subtracting) a quadratic term $~(\rho_1/\rho_2)\Delta^2$ as a "repelling" term. In fact, this is a great explanation of DSP-TT at the Objective Optimization view point. We in this paper analyze the problem on the other view point, which is to interpolate optimizations as dynamical systems. That is why we directly manipulate the dynamics (Equation 2), instead of adding a quadratic term in the objective. (Rigorously, $\rho_1$ is not a constant but a variable, so the above repelling explanation may not be fully equivalent to our representation. For instance, if using the modified loss function along, the algorithm does not know when an exit point is reached and stop the search process.)
> - Thank you for noting the references. We will add more recent papers regarding ensemble in the Reference in the next update. And we want to emphasize that our major contribution is to develop a systematic way of finding multiple local optimal solutions in training deep neural networks. And ensemble is one natural by-product. One can also pick the best individual network from the list. In Table 4 of the updated version, we include fully trained tier-1 network performances, showing that training from an exit point found by DSP-TT is better than from a random initialization.

---

> ### Author Response · Authors · 2020-11-18
> **Response to Reviewer 4 (part 2)**
>
> - We have also updated the manuscript with some recent papers[1,2,3,4] in Section 2.
> - Specifically for the two papers that you mentioned in the review. [1] improved the ensemble inference efficiency via a teacher-student paradigm distilling the knowledge of an ensemble into one single network. Since their focus is on improving the inference speed, a fully trained teacher ensemble is still required, with extra training overhead for student model as well as the discriminator models. [2] built ensembles by randomly initialize on a subparameter space, aiming to alleviate the exponentially growing number of local minima on deep networks. While their performance is not satisfying in Table 16, we believe their method of random initialization can be improved by our systematic DSP-TT method.
>
> [1] Zhiqiang Shen,  Zhankui He,  and Xiangyang Xue.   Meal:  Multi-model ensemble via adversarial learning. In AAAI, 2019.
> [2] Yongquan Yang, Haijun Lv, Ning Chen, Yang Wu, Jiayi Zheng, and Zhongxi Zheng. Local minima found  in  the  subparameter  space  can  be  effective  for  ensembles  of  deep  convolutional  neural networks. Pattern Recognition, 109(2021), August 2020.
> [3]Ruqi Zhang, Chunyuan Li, Jianyi Zhang, Changyou Chen, and Andrew Gordon Wilson.  Cyclical stochastic gradient mcmc for bayesian deep learning. In ICLR, 2020.
> [4] Stanislav Fort, Huiyi Hu, and Balaji Lakshminarayanan.  Deep ensembles:  A loss landscape perspective. arXiv:1912.02757, 2020.

---

### Official Review · AnonReviewer1 · 2020-10-28
**Interesting idea; good performance**

**Rating:** 6
**Confidence:** 4

**Review:**

This paper  proposes an intersting Dynamic Search Path TRUST-TECH training method for deep neural nets. In contrast to other global solvers, the proposed method efficiently explores the parameter space in a systematic way. Specifically, a Dynamic Searching Path method is proposed to make the original TRUST-TECH applicable to deep neural networks. Then,  the batch evaluation formula is used to increase the algorithm efficiency. Furthermore, the DSP-TT Ensembles is construed to  improve the model performance. I think the idea is interesting. The experimental results validate the effectiveness of the proposed method. The writing is good.

---

> ### Author Response · Authors · 2020-11-18
> **Response to Reviewer 1**
>
> Thank you very much for your comments and feedback. We appreciate your review, and we made the following major changes compared to the original version so far.
> - An ablation test on the DSP-TT hyperparameters $\rho_1$ and $\rho_2$ is added in Section 5.4.
> - The relations between LOS and SEP is clarified in Section 3.1.
> -  The derivation of the DSP method is simplified in Section 3.2.1
> - Table 4 is added to show the individual tier-1 solution (fully trained) performances.
> - Some clarification on definition of *high-quality* in Sec. 4.

---

### Official Review · AnonReviewer2 · 2020-10-29
**Method for applying to TRUST-TECH neural networks ensemble**

**Rating:** 5
**Confidence:** 3

**Review:**

This paper proposes a new method for applying the TRUST-TECH method to the ensemble of deep neural networks (DNNs). When applying TRUST-TECH to a deep neural network, it is difficult to determine the direction and exit point. This paper introduces Dynamic Searching Paths (DSP) to solve these problems. The proposed method can apply TRUST-TECH method to DNNs using Stochastic Gradient Descent (SGD) with minor memory overhead.

[+] Propose a method of applying TRUST-TECH to find local optimal solutions (LOS) of DNNs\
[+] Introduce DSP for exploration in high-dimensional parameter space\
[+] High ensemble performance through DSP-TT

[-] Claims not sufficiently proven or supported\
The paper repeatedly claims that the solution found by the proposed method is high quality and diversity. However, it is hard to find a theoretical and experimental basis for high-quality solutions. Indeed, the definition of a high-quality solution is also unclear. The following is the part that mentioned the quality of the local solution on page 1.
"A popular metric for the quality of a local solution is to measure its generalization capability, which shows the gap between the training and testing performances."
According to this explanation, high quality should have a small gap between training and testing performances, but none of the paper's experiments can support this.

Also, the relationship between diversity and performance is not convincing. The paper uses parameter distance and output correlation as measures for diversity. Tables 2 and 3 show that the relationship between these metrics and performance is very weak. In many cases, models with high output correlation have better performance, or models with close distances sometimes have higher performance. It can be seen from two aspects: these metrics are not suitable for measuring diversity, or the relationship between diversity and performance is weak.

[-] Unconvincing baseline results\
According to the FGE paper, ResNet-164 has errors of 20.2 and 4.54 with 150 epochs budget in CIFAR-100 and CIFAR-10, respectively, and 18.67 and 4.21 in 300 epochs. This is an almost similar result to DSP-TT with 200 epochs budget. However, in this paper, the FGE method uses 200 epochs and has errors of 20.75 and 4.67 in CIFAR 100 and CIFAR-10, respectively. Since it uses more budget, the performance should be higher, but it seems that FGE is not implemented properly. The explanation about this result is missing, making it difficult to trust the paper's experimental results.

[-] Minor comments\
The image quality in Figure 1 is poor, so it is difficult to identify text even if it is enlarged.

The proposed method is somewhat novel in the aspect of suggesting TRUST-TECH for an ensemble of DNNs. However, it is difficult to give a high score because the experimental results do not support

---

> ### Author Response · Authors · 2020-11-18
> **Response to Reviewer 2**
>
> Thank you for your detailed comments and feedback. We appreciate your review and have made changes in the paper to better address your concerns. We next address the individual points:
> - The generalization capability is commonly defined as the gap between training and testing performance [1]. Due to the high expressivity of deep neural networks with huge number of parameters, it is often observed that the training loss is often near zero (or 100\% training accuracy on training set of an image classification problem). In that case, the gap between training and testing accuracy becomes: (train accuracy)-(test accuracy)=1.0-(test accuracy)=(test error)
> That is why we only show the test error in our manuscript, since training accuracy are almost 100\%, and the test error can be a valid representative of the generalization capability. This is also seen from our referenced papers, for example FGE and SSE.
> - In this paper, for building ensembles with a limited budget, *high-quality* is emphasized more on the diversity among the collection of local optimal neural networks found to better serve the ensemble, in stead of on a single network. This claim is supported empirically by the numerical results presented in tables 1, 2, and 3. On the other hand, when training budget is less constrained, *high-quality* of each tier-1 solution is emphasized as having better test accuracy than the tier-0 network. This clarification is added to the updated manuscript. We also include the Table 4 in the updated manuscript, which shows individuals of of fully trained tier-1 networks performs at least as good as the tier-0 network. This also suggests that training from an exit point found by DSP-TT is better than from a random initialization. We have updated the discussion on the definition on high-quality in Section 4.
> - Diversity is crucial for ensemble. Our contribution is not developing new diversity metrics, but finding multiple local optimal deep nets that are also diverse, in a systematic manner. With these diverse nets, quality ensembles become a natural by-product. Correlation between Parameter Distance (PD) and Pearson Output Correlation is not expected to be high, otherwise (if they are fully dependent) only one metric would be enough. That's why we consider 3 factors for diversity: Accuracy, PD and OC. PD and OC are popular metrics for diversity of an ensemble used by many researchers, references are listed in 5.3 on page 7. Table 3 shows that SSE and FGE have lower PD, but higher OC. We do not judge the causality between PD and OC, but intuitively they are correlated to some extend. (for example, if two set of parameters are extremely close, the outputs should be nearly identical, but as they move farther apart, behavioral differences cannot be predicted.) Table 3 also shows that DSP-TT-0-1 and the naive ensembles have similar PD and OC, which are both more diverse than SSE and FGE. We are open to any other effective diversity measures, most of these measures would base on either the parameter space or output projection space. For the purpose of our paper, PD and OC, along with the test performance, are enough.
> - For SSE and FGE, we use the program provided by the authors on github
> (SSE: https://github.com/gaohuang/SnapshotEnsemble; FGE: https://github.com/timgaripov/dnn-mode-connectivity).
> and train from scratch using Pytorch 1.0 on a single GPU (Nvidia GTX-1080Ti). For FGE, we could not reproduce the FGE results of ResNet164 with 150 epochs of training budget. We report the best performance we could get which is to train FGE(ResNet164) for 200 epochs, our 200-epoch ResNet164-FGE performance similar to that of 150-epoch FGE recorded in the original FGE paper. There may be other environment software or hyperparameter differences between our reproduction and the original FGE paper, but we make sure that all of our experiments are conducted within the same environment, so that the comparison is fair.
> - Figure 1 is reproduced in the updated manuscript with higher quality.
>
> [1] Yann LeCun, Yoshua Bengio, and Geoffrey Hinton. Deep learning. Nature, 521(7553):436–444, 2015. doi: 10.1038/nature14539.

---

### Official Review · AnonReviewer3 · 2020-10-29
**Algorithm to explore the space around local minima**

**Rating:** 5
**Confidence:** 3

**Review:**

The authors propose a method to obtain multiple local optimal solutions around the existing one. The algorithm consists of moving away from the local minima along some random direction, but biasing the direction with a true gradient of the loss.
The solutions are then combined into a single result by taking them as an ensemble.

I quite like the idea behind the the algorithm. Combining several nearby solutions might help the overall performance of the method. However, I see several important problems with the presented method.

First, it is not clear how sensitive the algorithm is to the the learning rates \rho_1 and \rho_2. I believe that they would very much depend on the problem at hand and even the loss surface around the local function.

Second, I had trouble understanding the theoretical ideas behind. The paper is a bit hard to read and would benefit a lot from proofreading. For example, SEP is never formally defined. It says at the bottom of p.4 that "SEPs (i.e. nearby LOSs)". So what is the difference between SEP and LOS? It is not clear why the variable updates are written as a state dynamical system instead of regular iterations. Was that necessary in any way?  It is not well described how exactly TRUST-TECH works and finds LOS. How exaclty high-quality local optima defined and how is it different from other local minima?

---

> ### Author Response · Authors · 2020-11-18
> **Response to Reviewer 3**
>
> Thank you for your detailed comments and feedback. We appreciate your review and have made changes in the paper to better address your concerns. We next address the individual points:
> - As stated in Section 5, $\rho_1$ is set to increase 0.001 per iteration, and $\rho_2$ is chosen as 0.1$\times$ the initial learning rate. As shown in all experiments in Section 4, this set of hyperparameters is robust across different architectures and datasets, namely, MLP trained on MNIST; VGG, ResNet and DenseNet trained on CIFAR10 and CIFAR100. We also conducted a sensitivity test on the two parameters in Section 5.4 of the updated manuscript.
> - As discussed in 3.1, the TRUST-TECH methodology transforms the original optimization problem into a corresponding dynamical system, and therefore each Local Optimal Solution of the optimization problem is represented as an SEP in the dynamical system. In this regard, TRUST-TECH builds a bridge between optimization and dynamical systems, and therefore these terms are used interchangeably in the rest of this paper. The formal definition of Stable Equilibrium Point of a dynamical system is in reference [1], which is also included as a reference in our manuscript.
> - In this paper, for building ensembles with a limited budget, *high-quality* is emphasized more on the diversity among the collection of local optimal neural networks found to better serve the ensemble, in stead of on a single network. This claim is supported empirically by the numerical results presented in tables 1, 2, and 3. On the other hand, when training budget is less constrained, *high-quality* of each tier-1 solution is emphasized as having better test accuracy than the tier-0 network. This clarification is added to the updated manuscript. We also include the Table 4 in the updated manuscript, which shows individuals of of fully trained tier-1 networks performs at least as good as the tier-0 network. This also suggests that training from an exit point found by DSP-TT is better than from a random initialization.
>
> [1] Hsiao-Dong Chiang and Chia-Chi Chu. A systematic search method for obtaining multiple local op- timal solutions of non-linear programming problems. IEEE Transactions on Circuits and Systems I: Fundamental Theory and Applications, 43(2):99–109, 1996.

---

### Author Response · Authors · 2020-11-16
**General Comment**

We would like to appreciate the reviewers for their in-depth comments. We will address concerns for each reviewer respectively.

---

### Author Response · Authors · 2020-11-18
**Updates on the manuscript**

While detailed replies to each reviewer are conducted under each review, respectively, we here summarize major changes to our manuscript in order to better address the concerns from reviewers. They are summarized as follows.
- An ablation test on the DSP-TT hyperparameters $\rho_1$ and $\rho_2$ is added in Section 5.4.
- The relations between LOS and SEP is clarified in Section 3.1.
- The derivation of the DSP method is simplified in Section 3.2.1.
- Clarifications on definition of *high-quality* in Sec. 4.
- Table 4 is added to show the *high-quality* of individual tier-1 solution (fully trained) performances.
- Replace Figure 1 with a high quality image.
- Clarification on the *generalization capability* in Section 1.
- Clarification on code source of FGE and SSE in Table 2.
- Additional survey of recent papers regarding ensemble of deep neural networks in Section 2.

---

### Decision · Program_Chairs · 2021-01-07
**Final Decision**

**Decision:**

Reject

**Comment:**

This work proposes a method to discover neighboring local optima around an existing one. Reviewers all found the idea interesting but argued that the paper needed more work. In particular, some of the claims are too informal or not sufficiently supported and the reviewers found the key section were difficult to follow. The paper should be resubmitted after improving the presentation of the results.